# Acute Exposure to Aerosolized Nanoplastics Modulates Redox-Linked Immune Responses in Human Airway Epithelium

**DOI:** 10.3390/antiox14040424

**Published:** 2025-03-31

**Authors:** Joshua D. Breidenbach, Benjamin W. French, Upasana Shrestha, Zaneh K. Adya, R. Mark Wooten, Andrew M. Fribley, Deepak Malhotra, Steven T. Haller, David J. Kennedy

**Affiliations:** 1Department of Medicine, College of Medicine and Life Sciences, University of Toledo, Toledo, OH 43614, USAbenjamin.french2@rockets.utoledo.edu (B.W.F.); upasana.shrestha@rockets.utoledo.edu (U.S.); zaneh.adya@rockets.utoledo.edu (Z.K.A.); andrew.fribley@utoledo.edu (A.M.F.); deepak.malhotra@utoledo.edu (D.M.); 2Biochemistry and Biotechnology Group, Bioscience Division, Los Alamos National Laboratory, Los Alamos, NM 87545, USA; 3Department of Medical Microbiology and Immunology, College of Medicine and Life Sciences, University of Toledo, Toledo, OH 43614, USA; r.mark.wooten@rockets.utoledo.edu

**Keywords:** nanoplastics, aerosols, airway inflammation, human airway epithelium, environmental health

## Abstract

Micro- and nanoplastics (MPs and NPs) are pervasive environmental pollutants detected in aquatic ecosystems, with emerging evidence suggesting their presence in airborne particles generated by water body motion. Inhalation exposure to airborne MPs and NPs remains understudied despite documented links between occupational exposure to these particles and adverse respiratory outcomes, including airway inflammation, oxidative stress, and chronic respiratory diseases. This study explored the effects of acute NP exposure on a fully differentiated 3D human airway epithelial model derived from 14 healthy donors. Airway epithelium was exposed to aerosolized 50 nm polystyrene NPs at concentrations ranging from 2.5 to 2500 µg/mL for three minutes per day over three days. Functional assays revealed no significant alterations in tissue integrity, cell survival, mucociliary clearance, or cilia beat frequency, suggesting intact epithelial function post-exposure. However, cytokine and chemokine profiling identified a significant five-fold increase in CCL3 (MIP-1α), a neutrophilic chemoattractant, in NP-exposed samples compared to controls. This was corroborated by increased neutrophil chemotaxis in response to conditioned media from NP-exposed tissues, indicating a pro-inflammatory neutrophilic response. Conversely, levels of interleukins (IL-21, IL-2, IL-15), CXCL10, and TGF-β were significantly reduced, suggesting immunomodulatory effects that may impair adaptive immune responses and tissue repair mechanisms. These findings demonstrate that short-term exposure to NP-containing aerosols induces a distinct pro-inflammatory response in airway epithelium, characterized by enhanced neutrophil recruitment and reduced secretion of key immune modulators. These findings underscore the potential for aerosolized NPs to induce oxidative and inflammatory stress, raising concerns about their long-term impact on respiratory health and redox regulation.

## 1. Introduction

With ever-increasing global industrialization, the use of plastics has skyrocketed [1]. In 2016, an astounding 335 million tons of plastic was produced globally, with an estimated 12.7 million tons entering the ocean from land-based sources. This figure is expected to triple by the year 2050 [2,3]. Due to various physical, chemical, and biological processes, plastics eventually break down into smaller fragments, resulting in the formation of microplastics (MPs; 1 μm–5 mm) and nanoplastics (NPs; <1 μm) [1,4,5]. According to the Guidance on Monitoring of Marine Litter, microplastics can be categorized based on size, falling into the categories of 1–5 mm or 20 μm–1 mm. These microplastics inevitably break down into smaller particles, measuring less than 100 nm [6]. MPs and NPs with a wide range of polymer compositions (e.g., polypropylene (PP), polystyrene (PS), polyethylene (PE)) have been detected in both ocean and freshwater ecosystems, polluting these waters [7,8]. In addition to this, these plastic particles are transported over a wide distance via wind and atmospheric fallout [1,9].

Much research has been conducted to investigate the implications of MP and NP pollution on both local aquatic and terrestrial ecosystems, as well as on human health. Given the ubiquity and persistent nature of MPs and NPs in the environment, human exposure to these pollutants can occur via either ingestion, inhalation, or dermal contact [10]. Following exposure, MPs and NPs exhibit surface reactivity that may drive redox imbalances and exert deleterious toxicity to humans, including inflammation, cytotoxicity, and apoptosis in lungs [11,12,13]; intestinal barrier disruption [14]; increase in pro-inflammatory cytokines like interleukins (ILs) 6 (IL-6), and 1β (IL-1β) and tumor necrosis factor-α (TNFα) [15,16]; and cause oxidative stress, inflammation, and cytotoxicity on cerebral and epithelial human cell lines [17]. PS is among the “Big Five” plastics that dominate global plastic production and consumption, and several studies have linked PS-MPs and -NPs to the induction of inflammatory responses and cellular stress [18,19,20]. In addition to in vitro studies, there are multiple in vivo studies using mice models that present toxicities of MPs and NPs [10,21,22]. One study examined aerosolized PS particles at multiple concentrations for two weeks; the mice showed significant increases in inflammatory cytokines, histopathological changes, and the number of immune cells collected in the bronchoalveolar lavage fluid [16]. In a study involving MP-exposed mice, there was intestinal inflammation and gut microbiota dysbiosis [19]. In another study where mice were exposed to MPs, increased levels of inflammatory cytokines and chemokines and NLR family pyrin domain-containing 3 (NLRP3) inflammasome activation indicated significant lung injury [22].

Many of these studies are primarily focused on the oral route of exposure, with some considering the route of dermal contact to MPs and NPs. There has been limited research conducted into the inhalation of these plastic particles, though recent evidence suggests that natural water body motion is more than sufficient to create airborne water droplets that contain these plastic particles [5]. Additionally, other sources of MPs and NPs include synthetic textiles, sludge byproducts, tire erosion, and occupational exposures. Such exposures have been indicative of airway irritation, neutrophilic inflammation, translocation, increased risk of lung carcinoma and chronic respiratory diseases such as asthma, and even respiratory failure [1,23]. Moreover, MPs and NPs persist in the environment for years, potentially leading to accumulative effects on human health [24].

Considering the widespread distribution of airborne plastics, the current study set out to establish the impact of aerosolized NP exposure on airway epithelial cells using a three-dimensional human airway epithelial cell model. This model was established from a pool of 14 health donor cells and generated multiple layers of cells with functional cilia and mucus production. The culture utilized an air–liquid interface for aerosol exposure, allowing for direct contact of aerosol to tissue culture. Here, we tested the hypothesis that acute exposure to aerosolized NPs induces a pro-inflammatory effect in human airway epithelium. The effect of aerosolized NPs on the model was evaluated with cell survival and functional assays that examined tissue integrity, mucociliary clearance, and cilia beat frequency.

## 2. Materials and Methods

### 2.1. Microplastics

Polybead^®^ Microspheres 0.05 μm (polystyrene beads, 2.5% *w*/*v* in aqueous solution, 0.05 μm mean diameter, 3.64 × 10^14^ particles/mL, cat. 08691-10) were purchased from Polysciences (Warrington, PA, USA).

### 2.2. Primary Cell Culture and Exposure

A 3D model of cultured primary human airway epithelial cells was constructed from primary cells taken from 14 donors, acquired and provided by Epithelix with ethical approval and donor consent (MucilAir^TM^; Epithelix; Geneva, Switzerland). Donor cells were pooled to reduce variation introduced between donors. These cell cultures were established on 24-well transwell inserts at an air–liquid interface with serum-free MucilAir^TM^ Medium (Epithelix; Geneva, Switzerland). This model presents both functional beating cilia and mucus secretion.

The reconstructed primary 3D airway epithelium was subject to exposure 3x/day for 3 days to either polystyrene plastic beads (0.05 µM mean diameter; Polysciences Inc., Warrington, PA, USA) or vehicle (saline) using the Vitrocell-Cloud nebulization system from Epithelix. This system is designed for dose-controlled and spatially uniform aerosolization of liquid samples on cells cultured with air–liquid interfaces. Each aerosolization exposure lasted approximately 3 min. An estimated 0.43 μL/insert of solution was deposited during each exposure, based on the surface area of the 24-well inserts (0.33 cm^2^), the surface area of the Vitrocell-Cloud system (136.5 cm^2^), the 200 μL of solution nebulized per exposure, and an estimated 90% delivery efficacy.

### 2.3. Tissue Integrity

Tissue integrity was assessed by the transepithelial electrical resistance (TEER) assay. This assay measures electrical resistance via electrodes on both the apical and basal surfaces of the epithelial tissue inserts. As tissues are not as conductive as the electrodes, in-tact tissue presents resistance to the flow of electricity from one side to the other. Therefore, a loss of this resistance indicates a decrease in tissue integrity as there are fewer cells to form a continuous intercellular barrier via intercellular adhesion complexes. Functional epithelium tissues typically present 200–600 Ω × cm^2^, while deteriorated tissue (broken cellular junctions) measure under 100 Ω × cm^2^.

To conduct these measurements, 200 μL of normal saline was added to the apical compartment of MucilAir^TM^ cultures and resistance was subsequently measured using an EVOMX volt-ohm-meter (World Precision Instruments UK, Stevenage, UK) for each tested condition. Resistance values (Ω) were converted to TEER (Ω × cm^2^) using the following formula, TEER (Ω × cm^2^) = (resistance value (Ω) − 100(Ω)) × 0.33 (cm^2^), where 100 Ω is the resistance of the membrane and 0.33 cm^2^ is the total surface of the epithelium.

### 2.4. Cytotoxicity Assay

Cytotoxicity in the tissue inserts was tested by the lactate dehydrogenase (LDH) assay, measuring levels of LDH in conditioned cell cultured media. As this assay assumes that LDH is a stable enzyme primarily retained within intact cells, increased levels of LDH in conditioned media indicates cell death, and loss of integrity, thereby releasing LDH into the media. On the third day of this study, 24 h after the third and final exposure, 100 μL of basolateral medium was removed from each culture and incubated with the reaction mixture of the LDH Assay Kit-WST (Dojindo; Rockville, MD, USA; ref. CK12-20) following the manufacturer’s instructions. Quantification of LDH in each sample was measured using a microplate reader at 490 nm. To determine the percentage of cytotoxicity, the following equation was used:$$Cell\;Survival\;\left(X\right)=100(\%)-\left(\frac{A_{sample}-A_{low\;control}}{A_{high\;control}-A_{low\;control}}\right)*100(\%)$$
where *A* = absorbance values, and the high control value was obtained by 10% Triton X-100 apical treatment. The negative controls (vehicle) typically show a low daily basal LDH release, which is due to a physiological cell turnover in the model.

### 2.5. Cilia Beat Frequency and Mucociliary Clearance

Cilia beat frequency (CBF) and mucociliary clearance (MCC) were assessed by the movement of polystyrene beads along the apical surface of the epithelial tissue insert. Cilia beat frequency was measured via a three-part dedicated setup for this purpose. This system consisted of a camera-equipped microscope, a PCI card, and Epithelix’s Cilia-X CBF analysis software (https://serc.carleton.edu/introgeo/teachingwdata/Fourier.html, accessed on 10 February 2025). CBF is expressed in Hz (cycles or repetitions per second). Videos were generated at 125 frames per second, from which CBF was calculated by Epithelix software. MCC was monitored using a Sony XCD-U100CR camera (Sony; Geneva, Switzerland) connected to an Axiovert 200 M microscope (Zeiss; Oberkochen, Germany) with a 5× objective. Microbead movements were video-tracked at 2 frames per second for 30 images at room temperature. Three videos were taken per insert. Average bead movement velocities (μm/s) were calculated with ImageProPlus 6.0 software.

### 2.6. Protein Secretion

Before each dose, and 24 h after each exposure (plastics or saline), the primary human airway epithelial cell culture media was refreshed. Conditioned media taken from the culture post-exposure were frozen for analysis. To analyze the secreted proteins, the conditioned medium samples from each day were thawed and pooled within groups. Quansys Biosciences (Logan, UT, USA) performed protein measurements via custom multiplex ELISA-based Q-Plex^TM^ technology for quantification of 36 human cytokines and chemokines (Appendix A).

### 2.7. Chemotaxis

To assess the chemotactic potential of the conditioned epithelial medium sample, we performed Boyden chamber assays. Briefly, the Boyden chamber assay is conducted in the 96-well format using HTS Transwell insert plates with 3 µm pores in polycarbonate membranes (Corning, Cat: 3385; Corning, NY, USA). These HTS Transwell insert plates were coated with 50 µg/mL Matrigel (Corning, Cat: 356234; Corning, NY, USA) on each side in RPMI (Caisson Labs, RPL04—phenol red free; Smithfield, UT, USA) + 10% FBS (Gibco, Cat; 26140079; Waltham, MA, USA) at 37 °C for 30 min, followed by 30 min at room temperature. After coating, the plates were placed in a reservoir plates containing 200 uL of dBPS (Caisson Labs, PBL01—without calcium and magnesium; Smithfield, UT) per well while primary human neutrophils (Astarte Biologics/Cellero, Cat: 1025; Lowell, MA, USA) were placed at 1000 cells per well in RPMI + 0.5% FBS above the membrane within the transwell insert. A separate reservoir plate was used for the negative and positive controls (RPMI + 0.5% FBS and 100 nM fMLP (Sigma-Aldrich, F3506; Merck KGaA, Darmstadt, Germany) in RPMI + 0.5% FBS, respectively), as well as potentially attractant conditioned media (conditioned media diluted 1:2 in RPMI + 0.5% FBS). The insert plate was allowed to incubate for 30 min at room temperature before being carefully transferred to the experimental reservoir plate for a 12 h incubation at 37 °C. Chemotaxis was monitored for the entire incubation step on 1 h intervals using an Incucyte S3 (Sartorius AG; Göttingen, Germany) live cell imaging system. Migration from the top chamber to the bottom chamber was noted as a loss of total area of neutrophils on the top portion of the membrane.

### 2.8. Statistics

All assay-related data were processed using GraphPad Prism version 7.0.5 for Windows (GraphPad Software, San Diego, CA, USA) using Student’s *t*-test. Multiple groups were tested by ANOVA with Tukey’s multiple comparisons test, followed by multiple t-tests. All the data are expressed as the mean ± standard error of mean (SEM). Significance is marked as a *p*-value ≤ 0.05. *p*-value ranges are marked on figures as *, **, ***, or **** for *p*-values ≤ 0.05, 0.01, 0.001, and 0.0001, respectively.

## 3. Results

### 3.1. Effects of Aerosolized Nanoplastics Exposure on Functional Epithelial Measures

Tissue integrity was measured using the TEER assay, and interestingly, treatment with the aerosolized NPs (ANPs) at 2.5 µg/mL showed a significant increase in electrical resistance. This trend appears to continue at 250 and 2500 µg/mL but did not reach significance in either case (*n* = 3, *p*-value = 0.0997 and 0.1690, respectively) (Figure 1B). However, this value is within normal TEER values (between 200 and 600 Ω × cm^2^) of intact epithelium. There was a significant decrease in electrical resistance upon exposure to Triton X-100, indicating severe destruction of the airway epithelial cell barrier integrity.

The LDH assay was used to assess the cytotoxicity imposed by ANPs exposure by measuring levels in the conditioned cell culture media. The acute, repeated exposures to nanoplastic aerosol showed no cytotoxicity, even up to 2500 µg/mL (*n* = 3, Figure 1C).

CBF is one of several factors that influences the MCC, which aids in maintaining homeostasis and functions as a physical barrier for the immune system. Treatment with ANPs had a significant impact on the cilia beat frequency at every dose but did not appear to be dose-dependent (*n* = 3, Figure 1D). Isoproterenol was used as a positive control, being known to significantly increase CBF in many mammals, including humans [25,26]. Vehicle-induced CBF had an average of 3.918 Hz (±0.143 SEM); isoproterenol-induced CBF had an average CBF of 12.42 Hz (±0.104 SEM). At each dose of ANP exposure, 2.5 µg/mL, 250 µg/mL, and 2500 µg/mL, the average CBFs were 5.842 Hz (±0.111 SEM), 6.285 Hz (±0.224 SEM), and 5.634 Hz (±0.049 SEM), respectively. Treatment with ANPs resulted in a dose-dependent reduction in MCC, although the change was not significant, as determined by tracking the movement of beads across the cells (*n* = 3, Figure 1E).

### 3.2. Differential Secretion of Cytokine and Chemokine Proteins Following Aerosolized Nanoplastics Exposure

Based on the ELISA-based Q-Plex^TM^ quantification of secreted proteins, Figure 2 shows the six cytokines and chemokines observed to be significantly differentially secreted after exposure to aerosolized NPs. Secretion of macrophage inflammatory protein-1 alpha (MIP-1α) was significantly higher following ANP exposure, with an average of 4.94 pg/mL compared to vehicle (average 1 pg/mL). The observed increase in MIP-1α and neutrophilic chemotaxis indicates a pro-inflammatory milieu, potentially driven by oxidative stress induced by ANP exposure. On the other hand, ILs 21, 2, and 15 as well as C-X-C motif chemokine ligand 10 (CXCL10) and transforming growth factor β (TGF-β) were significantly decreased after exposure to aerosolized NPs (*n* = 3, Figure 2B–F). These significant reductions suggest an impairment of adaptive immune responses and redox-mediated tissue repair mechanisms.

### 3.3. Enhanced Neutrophilic Chemotaxis Following Aerosolized Nanoplastics Exposure

Interestingly, the vehicle showed higher neutrophilic chemotactic potential than RPMI alone. Beyond the chemotactic potential of the vehicle treatment, conditioned media from NP-exposed cells shows significantly increased chemotactic potential. The conditioned media showed significant increases in neutrophil migration at 10, 14, 16, and 18 h, suggesting an inflammatory response (*n* = 3, Figure 3A,B). There was a general trend of higher migration than both the vehicle treatment and unconditioned RPMI that did not reach significance outside of those time points.

## 4. Discussion

Plastic pollution has emerged as a significant global issue, with projections suggesting that by 2050, approximately 12 billion metric tons of plastic wastes will accumulate in landfills or in the natural environment [27]. Large plastic debris undergo fragmentation to form MPs and NPs [28]. In addition to the intentional presence of NPs in various personal care products such as shampoos, facial cleansers, and scrubs, the inevitable breakdown of plastics through physical, chemical, and biological processes also generates NPs [29,30]. NPs have been linked to detrimental health effects on humans [31], including inflammation and damage to the respiratory system [12,32]. In this context, we reported several impacts of aerosolized nanoplastic exposure in primary airway epithelial cells in a 3D cell culture model. Here, we used PS beads as our nanoplastics of choice for different reasons. PS is one of the most widely used plastics found in a diverse array of everyday items, including water bottles, food containers, food packaging, electrical and electronic equipment, and office supplies. Multiple studies have reported PS as among the most abundant plastic particles found in atmospheric deposition, highlighting the potential risks posed by PS particles via airborne exposure [5,33,34]. Moreover, PS-NPs, due to their small yet varying sizes, shapes, and surface charges, can readily traverse the food web, ultimately posing risks to human health [35,36,37].

We utilized a 3D model of cultured primary human airway epithelial cells to investigate the toxicity of ANPs by exposing the epithelial cells to PS-NPs by aerosolization. This study is limited in the variability of NP particle sizes tested and the lack of in vivo exposures conducted. However, the results obtained from human primary 3D cell culture methods are applicable to human health outcomes. This model represents a physiologically relevant model of the human airway, comprising basal cells, mucus-secreting goblet cells, and ciliated cells, showing cilia beating and mucociliary clearance [38,39]. This model has been employed in studies to assess the respiratory toxicity of inhaled drugs targeting respiratory diseases [38] and has been shown to express xenobiotic metabolism enzymes [40]. The single NP size selected does not represent the entire spectrum of nanoplastic particles that become aerosolized, but the 50 nm size is a common model particle size that may have higher uptake in cells [41,42,43].

As shown by the assessments of tissue function, acute exposure to ANPs does not have a significant impact on tissue integrity, cell survival, or mucociliary clearance. Interestingly, the acute ANP exposure did cause an increase in the CBF without a change in MCC, which may suggest nanoplastics negatively impact cilia beat efficiency. Normal CBF is essential for maintaining healthy respiratory functions [44]. CBF is used as an indicator of ciliary activity to evaluate the ciliotoxicity of different compounds upon nasal delivery [45]. Elevated CBF can be indicative of a stress response of the airway epithelium to perceived threats. This could be a protective mechanism, aiming to enhance the clearance of potentially harmful substances [46], such as ANPs in this context. Ciliary beating is a crucial component of the respiratory defense system, as the cilia drive the propulsion of airway surface liquid, which entraps airborne particles, including debris and microorganisms, and directs them toward the oropharynx for clearance via swallowing or coughing them out [47,48]. However, it is crucial to note that the prolonged exposure to ANPs and a persistent increase in CBF could lead to an imbalance in airway surface liquid composition, potentially leading to ciliary fatigue and impairing long-term respiratory epithelial function [49,50]. Such disruptions could have broader implications for airway clearance and host defense mechanisms.

Despite the lack of significant changes in some functional tissue measurements, there are several notable changes in cytokine and chemokine signaling after acute exposure to aerosolized nanoplastics. Amongst the cytokines and chemokines tested, there were six cytokines whose secretions were significantly impacted. The immunomodulatory effects observed in this study may be linked to disruptions in cellular redox signaling, as evidenced by increased pro-inflammatory markers and diminished reparative cytokines. These results align with the hypothesis that nanoplastics contribute to oxidative stress, driving inflammatory and immune dysregulation in the respiratory epithelium. Levels of secreted MIP-1α, a potent neutrophilic chemoattractant, were roughly five-fold higher in the ANP-exposed airway epithelia compared to the control. Elevated levels of MIP-1α are linked not only to inflammation but also to different lung diseases, including asthma, acute respiratory distress syndrome, and pulmonary fibrosis [51,52]. Moreover, overexpression of MIP-1α can amplify neutrophil recruitment, which potentially increases reactive oxygen species (ROS) production and oxidative stress in the airway epithelium [51]. Such ROS production by inflammatory cells and airway epithelial cells disrupt redox homeostasis and activates signaling pathways like NF-κB, which exacerbate inflammation [53,54].

Further evidence for nanoplastics driving a neutrophilic-based inflammatory response comes from the chemotaxis assay. Neutrophils are the most abundant leukocytes in the blood and are rapidly mobilized as a form of initial immune response to combat infections or repair tissue damage [55]. Neutrophils have been implicated in various respiratory disease, such as asthma, chronic obstructive pulmonary disease (COPD), and pulmonary fibrosis [56]. An elevated neutrophil migration rate could be indicative of an active immune response or a pathological condition like COPD [57,58]. Higher neutrophil migration at several time points (10, 14, 16, and 18 h), combined with the increased secretion of MIP-1α provide strong evidence of a neutrophilic-based inflammatory response induced by ANP exposure. These results are in agreement with occupational exposure data that have suggested neutrophilic inflammation as an outcome of aerosolized nanoplastics [59,60]. Neutrophils recruited to sites of inflammation undergo oxidative bursts, producing large amounts of ROS [61]. The excessive ROS production by neutrophils overwhelms epithelial antioxidant defenses, causing redox imbalance, disrupting homeostasis, and contributing to oxidative damage in the airway epithelium, as seen in inflammatory lung conditions [53,62]. While this could be a concern for anyone exposed to ANPs, it is possible that this effect would be amplified in those with chronic neutrophilic asthma and COPD.

The levels of five signaling molecules (IL-21, IL-15, IL-2, CXCL10, and TGFβ) were detected at significantly lower concentrations in the conditioned media. It should be noted that these signaling molecules share several common features in the context of immune regulation and respiratory infections, including intercellular signaling, enhancing Natural Killer (NK) cell functions, and influencing T cell responses [63,64,65]. While they each play a unique role in the immune system, their roles are often heavily interconnected [66]. As such, a reduction in these cytokines following nanoplastic exposure in the airway epithelium has several important implications for immune function and tissue health. Interleukins modulate innate and adaptive immunity, as they coordinate the various T cells (IL-21, -15, -2), B cells (IL-21), and NK cells (IL-21, -15, -2) present in tissues [67,68,69,70,71,72]. Reductions to these interleukins, which are critical for respiratory immunity, is linked to reduced immune cell recruitment to the lungs and could compromise the immune system’s ability to combat pathogens and maintain lung immune surveillance [67,71,72]. Even more, reduction in these key cytokines results in reduced antioxidant defense systems, which leaves lung cells more vulnerable to ROS-induced damage [73]. As an example, a decrease in IL-21 can impact mitochondrial biogenesis and function, which can lead to increased ROS production and further cellular damage [74]. Similarly, CXCL10 is vital for protective immune responses, as it facilitates recruitment and activation of NK cells while also attracting activated T cells, NK cells, and dendritic cells to infection sites [75,76]. Lower CXCL10 levels may potentially result in reduced recruitment of these immune cells to the airway epithelium, leading to an inadequate immune response to inflammation or infections, thereby prolonging the oxidative stress triggered by nano- and microplastics. TGF-β is essential in maintaining lung homeostasis, regulating immune responses, maintaining epithelial integrity, as well as the control of normal tissue repair following lung injury [77]. In addition, it helps regulate immune responses to pulmonary infections, potentially limiting excessive inflammation [78]. Decreased TGF-β could impair epithelial repair and regeneration after damage, hindering tissue healing. Further, TGF-β has a critical role in regulation of oxidative stress [79]. Downregulation of TGF-β may lead to mitochondrial dysfunction, reduced antioxidant defenses, and increased ROS production [80]. However, reduced TGF-β could also reduce fibrosis [81], which may be a protective mechanism in the context of microplastic exposure.

Overall, the reduction in these secreted cytokines and chemokines indicates that nanoplastics play an immunomodulatory effect in the airways, driving a neutrophilic inflammatory response and an oxidative outburst while suppressing key signals that would otherwise trigger robust innate and adaptive immune responses. Micro- and nanoplastics have emerged as significant environmental stressors, and similarly to these tiny plastic particles, other environmental stressors, such as particulate matter and harmful algal bloom (HAB) toxins, contribute to human airway inflammation, redox imbalances, and injury [82,83,84]. Chronic exposure to these airborne pollutants may impair the lung’s natural antioxidant defense systems, such as glutathione (GSH), leading to redox imbalances [85,86]. Therefore, these findings underscore the potential impact of micro- and nanoplastic exposure on respiratory immune function and tissue health.

## 5. Conclusions

Using ex vivo experiments, we reported that acute exposure to aerosolized nanoplastics induces weakened and dysregulated immune responses in primary airway epithelial cells. This potentially disrupts the balance between different immune cell types and weakens antioxidant defenses, leaving our airway epithelia more vulnerable to ROS-induced damage. Consequently, this affects the long-term ability of the airway epithelium to cope with the environmental stressors like microplastics and HAB aerosol exposure. Due to high (and increasing) levels of pollution and the persistence of plastics in the environment, aerosolized plastic particles will become increasingly relevant in the years to come, and therefore increasingly concerning for global human health. Additional research to investigate the potential risks of ANPs on different human organ systems is important to prompt the development of stringent guidelines and regulations regarding the use of plastics in daily lives.

## Figures and Tables

**Figure 1 antioxidants-14-00424-f001:**
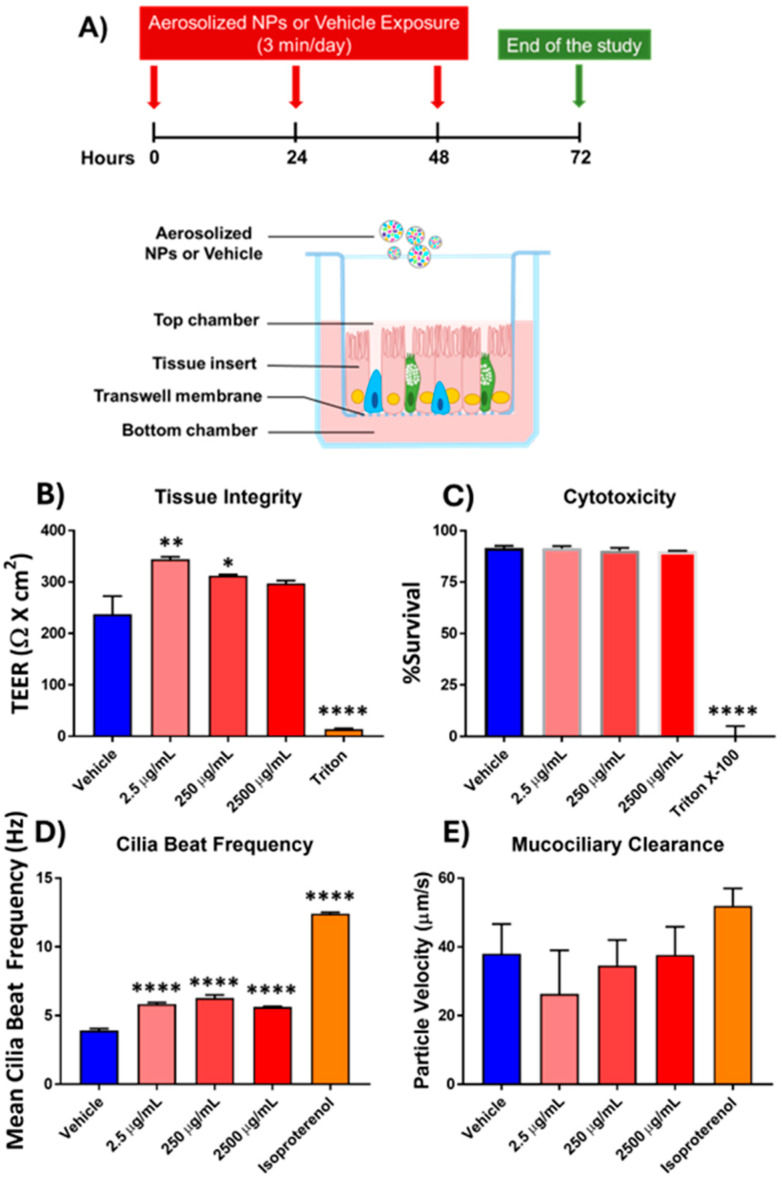
Measurements of functional tissue changes following aerosolized nanoplastic exposure. (**A**) Schematic of 3D airway epithelial tissue inserts in transwell culture conditions and timeline of exposure. (**B**) Tissue integrity—TEER of insert wells measured after exposure to aerosolized NPs. Values are displayed as the resistance in ohms multiplied by the area of the tissue insert in cm^2^. (**C**) Cytotoxicity—LDH in the epithelium conditioned media measured by LDH assay normalized against vehicle (100% survival). Values are displayed as percent of the average amount recovered from the samples exposed to the positive control (Triton X-100). (**D**) Cilia beat frequency—the frequency of cilia movement at the apical surface measured by video analysis via custom software. Values are displayed in hertz (Hz). (**E**) Mucociliary clearance—the net movement caused by cilia at the apical surface measured by high frame-rate videos tracking the movement of beads across the cells within the insert. Values are displayed as mean particle velocity in µm/s. All experiments shown contain *n* = 3/group. For all the measurements, significance by ANOVA followed by *t*-tests, where *, **, and **** correspond to *p*-values ≤ 0.05, 0.01, and 0.0001, respectively.

**Figure 2 antioxidants-14-00424-f002:**
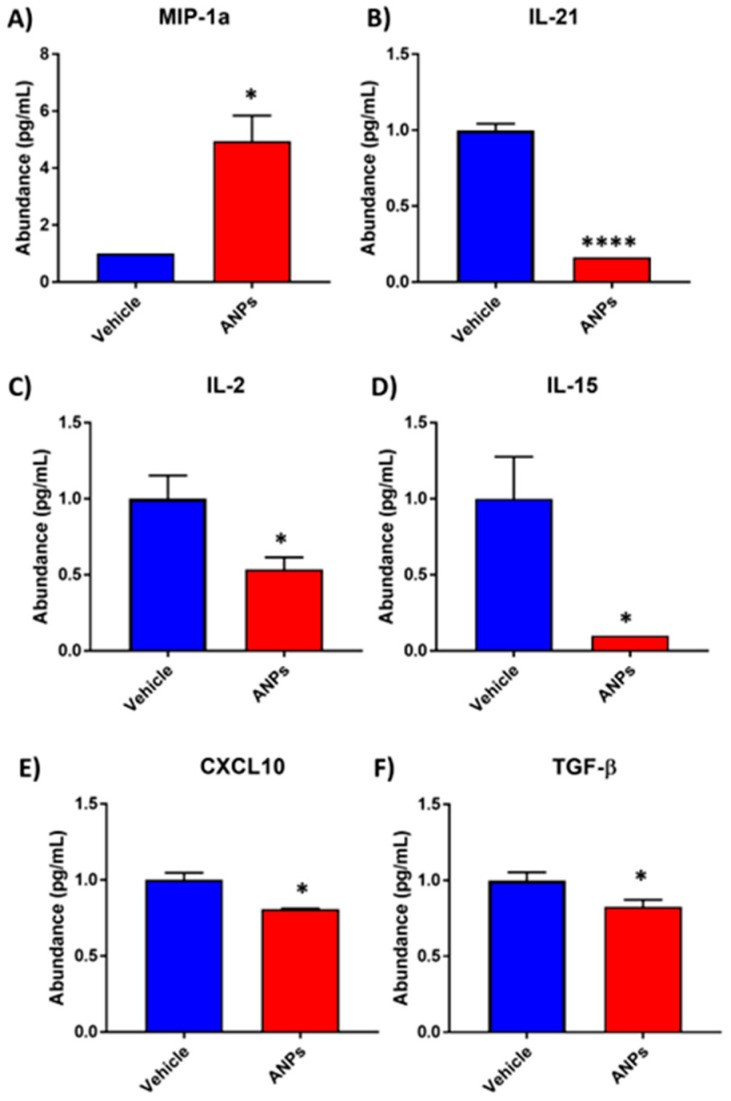
ELISA-based Q-Plex^TM^ Quantification of secreted proteins in conditioned media post-exposure to aerosolized nanoplastic (ANP) particles. Concentration of cytokine and chemokine proteins (**A**) MIP-1α, (**B**) IL-21, (**C**) IL-2, (**D**) IL-15, (**E**) CXCL10, and (**F**) TGF-β in the conditioned cell culture media recovered from tissue inserts exposed to ANPs or vehicle was determined. Values were normalized by total estimated protein and displayed as pg/mL. All experiments contain *n* = 3/group. Significance by Student’s *t*-test where * and **** correspond to *p*-values ≤ 0.05 and 0.0001, respectively.

**Figure 3 antioxidants-14-00424-f003:**
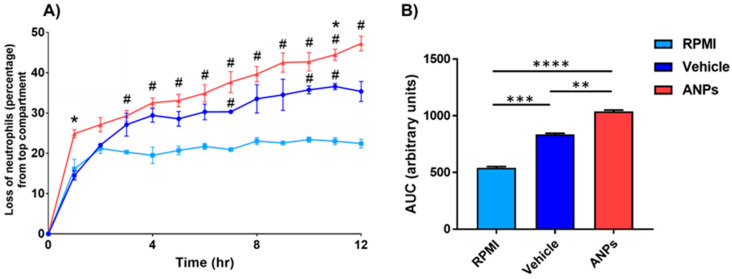
Chemotactic potential of neutrophils in conditioned cell culture media of aerosolized nanoplastic (ANP)-exposed cells. (**A**) Readings were taken at each hour post-exposure and were normalized against the initial surface area covered by neutrophils at 0 h. Loss of neutrophil area indicates higher chemotactic potential. (**B**) Total area under the curve (AUC) of neutrophil area loss at 12 h. All experiments contain *n* = 3/time point. Significance was determined by multiple t-tests (**A**) or a one-way ANOVA followed by *t*-tests (**B**) where *, **, ***, and **** correspond to *p*-values ≤ 0.05, 0.01, 0.001, and 0.0001, respectively. Time points marked with # indicate a difference from RPMI alone.

## Data Availability

Data are contained within the article or Appendix A. Further inquiries may be directed at the corresponding authors.

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
