# Peer review of "Acute Exposure to Aerosolized Nanoplastics Modulates Redox-Linked Immune Responses in Human Airway Epithelium"

_antioxidants, 2025, doi:10.3390/antiox14040424_

Round 1

Reviewer 1 Report

In this manuscript, Breidenbach et al. study the effects of aerosolized nanoplastics (ANPs) on airway epithelial cells using a 3D cell culture model. The topic is relevant, and the use of a 3D culture system is a strength. The findings on changes in cytokine secretion and neutrophilic chemotaxis are interesting and suggest a potential pro-inflammatory response caused by ANPs. However, the study has several weaknesses that limit its overall contribution.

1) The study does not provide much new insight into how nanoparticles affect airway epithelial cells. It focuses on changes in cytokines and chemotaxis but does not explore the pathways or mechanisms involved. In particular, there is no evidence showing how oxidative stress might play a role in these responses, which is an important aspect of nanoparticle toxicity.

2) Since this is being submitted to an oxidative stress-focused journal, it is a major issue that there is no direct evidence of oxidative stress. The study does not measure reactive oxygen species (ROS), antioxidant activity, or any other indicators of oxidative stress.

3) The experiments are also limited to a 3D cell culture model, which, while useful, cannot fully replicate the complexity of in vivo systems. The observation period is short (three days), and only one size of nanoplastics (50 nm) was tested. 

4) while changes in cytokines and chemotaxis were observed, the study does not show the functional impact of these findings.

1) In addition to the Student’s t-test, ANOVA with appropriate post hoc analysis should be described.

2) The methods for the 3D culture should be described in greater detail. Specifically, it is important to clarify how the samples were obtained, how multiple samples were handled within a single culture, and whether proper permissions or ethical approvals were applied.

3) The number of experiments (n) should be clearly stated.

4) the meaning of different colored lines in Fig. 3A should be indicated in legend or inside the graph.

Author Response

Comment 1:The study does not provide much new insight into how nanoparticles affect airway epithelial cells. It focuses on changes in cytokines and chemotaxis but does not explore the pathways or mechanisms involved. In particular, there is no evidence showing how oxidative stress might play a role in these responses, which is an important aspect of nanoparticle toxicity.

Response 1: Thank you for the feedback! We have now acknowledged the limitation of our study in the Discussion section: “We utilized 3D model of cultured primary human airway epithelial cells to investigate the toxicity of ANPs by exposing the epithelial cells to PS-NPs by aerosolization. This study is limited in the variability of NP particle sizes tested and the lack of in vivo exposures conducted. However, the results obtained from human primary 3D cell culture methods are applicable to human health outcomes. This model represents a physiologically relevant model of the human airway, comprising basal cells, mucus-secreting goblet cells, and ciliated cells, showing cilia beating and Mucociliary clearance [38, 39]. This model has been employed in studies to assess the respiratory toxicity of inhaled drugs targeting respiratory diseases [38] and has shown to express xenobiotic metabolism enzymes [40]. The single NP particle size selected does not represent the entire spectrum of nanoplastic particles that become aerosolized, but the 50 nm size is a common model particle size that may have higher uptake in cells [41-43].”

We would also like to emphasize that our study utilizes a 3D cell culture system with human primary cells and incorporates a key immune cell type (neutrophils) to perform functional assays (Figure 3), highlighting the importance of cell-cell interactions.

Comment 2: Since this is being submitted to an oxidative stress-focused journal, it is a major issue that there is no direct evidence of oxidative stress. The study does not measure reactive oxygen species (ROS), antioxidant activity, or any other indicators of oxidative stress.
Response 2: Thank you for your feedback! W
e have exhausted experimental material and do not have the funding necessary to get fresh samples for additional tests.

Comment 3: The experiments are also limited to a 3D cell culture model, which, while useful, cannot fully replicate the complexity of in vivo systems. The observation period is short (three days), and only one size of nanoplastics (50 nm) was tested.
Response 3: Thank you for the feedback! Our goal was to study the effects of acute aerosolized exposure, as outlined in the title of our paper. We have updated the hypothesis statement within the introduction to now read “Here, we tested the hypothesis that acute exposure to aerosolized NPs induces a pro-inflammatory effect in human airway epithelium.”

We have also now included the limitations of this study in the Discussion section: “We utilized 3D model of cultured primary human airway epithelial cells to investigate the toxicity of ANPs by exposing the epithelial cells to PS-NPs by aerosolization. This study is limited in the variability of NP particle sizes tested and the lack of in vivo exposures conducted. However, the results obtained from human primary 3D cell culture methods are applicable to human health outcomes. This model represents a physiologically relevant model of the human airway, comprising basal cells, mucus-secreting goblet cells, and ciliated cells, showing cilia beating and Mucociliary clearance [38, 39]. This model has been employed in studies to assess the respiratory toxicity of inhaled drugs targeting respiratory diseases [38] and has shown to express xenobiotic metabolism enzymes [40]. The single NP particle size selected does not represent the entire spectrum of nanoplastic particles that become aerosolized, but the 50 nm size is a common model particle size that may have higher uptake in cells [41-43].”

Comment 4: while changes in cytokines and chemotaxis were observed, the study does not show the functional impact of these findings.
Response 4: Thank you for your comment! In our results section (Figure 1), we show four functional measures that were not significantly modified by the acute exposures: Tissue integrity, as measure by electrical resistance; cell survival, measured via the LDH assay; as well as Cilia Beat Frequency and Mucociliary Clearance measured by advanced imaging and computational software.
We have modified our discussion to better discuss the functional changes: “As shown by the assessments of tissue function, acute exposure to ANP particles does not have a significant impact on tissue integrity, cell survival, or mucociliary clearance. Interestingly, the acute ANP exposure did cause an increase in the CBF without a change in MCC, which may suggest nanoplastics negatively impact cilia beat efficiency. CBF is used as an indicator of ciliary activity to evaluate the ciliotoxi-city of different compounds upon nasal delivery [41]. Normal CBF is essential for maintaining healthy respiratory functions [42]. CBF is used as an indicator of ciliary activity to evaluate the ciliotoxicity of different compounds upon nasal delivery [41].  Elevated CBF can be indicative of a stress response of the airway epithelium to perceived threats. This could be a protective mechanism, aiming to enhance the clearance of potentially harmful substances [43], such as ANPs in this context. Ciliary beating is a crucial component of respiratory defense system, as the cilia drives the propulsion of airway surface liquid, which entraps airborne particles, including debris and microorganisms, and directs them toward the oropharynx for clearance via swal-lowing or coughing them out [44, 45]. However, it is crucial to note that the prolonged exposure to ANPs and a persistent increase in CBF could lead to an imbalance in air-way surface liquid composition, potentially leading to ciliary fatigue and impairing long-term respiratory epithelial function [46, 47]. Such disruptions could have broader implications for airway clearance and host defense mechanisms.”

Comment 5: In addition to the Student’s t-test, ANOVA with appropriate post hoc analysis should be described.
Response 5: Thank you for the feedback! We have updated our Statistics section to include a description of our ANOVA: “All assay-related data were processed using GraphPad Prism version 7.0.5 for Windows (GraphPad Software, San Diego, California, USA, www.graphpad.com) using Student’s t-test. Multiple groups were tested by ANOVA with Tukey's multiple comparisons test, followed by multiple t tests. All the data were expressed as the mean ± standard error of mean (SEM). Significance is marked as a p-value ≤ 0.05. P-value ranges are marked on figures as *, **, ***, or **** for p-values ≤ 0.05, 0.01, 0.001, and 0.0001 respectively.” 

Comment 6: The methods for the 3D culture should be described in greater detail. Specifically, it is important to clarify how the samples were obtained, how multiple samples were handled within a single culture, and whether proper permissions or ethical approvals were applied.
Response 6: Thank you for the feedback! We did not collect the cells ourselves. The MucilAir system is constructed by Epithelix, where they handle the patient selection and sample collection. These are commercially available systems made with ethical approval and donor consent (https://www.epithelix.com/products/mucilair). The cells from all donors were mixed thoroughly and cultured from that mixture to minimize the influence of donor variability. 
We have amended the text to now read: “A 3D model of cultured primary human airway epithelial cells was constructed from primary cells taken from 14 donors, acquired and provided by Epithelix with ethical approval and donor consent (MucilAir™; Epithelix; Geneva, Switzerland).”

Comment 7: The number of experiments (n) should be clearly stated.
Response 7: We thank the reviewer for their feedback. We have added the (n) value for all experiments to the body of the text.

Comment 8: The meaning of different colored lines in Fig. 3A should be indicated in legend or inside the graph.
Response 8: Thank you for the feedback. We have updated Figure 3 to include a legend.

Reviewer 2 Report

None.

To the authors:

The original manuscript describes the effect of polystyrene microplastics and nanoplastics in the airway epithelium response (derived from 14 healthy donors) using functional tests. The manuscript is very novel and shows very interesting results. I just found minor comments that I think would improve the quality and transparency of the results.

General comments: minor

  1. Line 42. We are already in 2025, so please amend the sentence.
  2. Line 63. In vitro and in vivo do in italics.
  3. Line 212. Define SEM before use it.
  4. For bar graphs, show the experimental values (Figure 1B, 2, and 3B).
  5. Please revise this article and include in your study: doi: 10.1016/j.intimp.2024.113921.

Author Response

Comment 1: Is it necessary to include study limitations in the discussion? Yes

Response 1: Thank you for the comment! We have modified our Discussion to include limitations: “We utilized 3D model of cultured primary human airway epithelial cells to investigate the toxicity of ANPs by exposing the epithelial cells to PS-NPs by aerosolization. This study is limited in the variability of NP particle sizes tested and the lack of in vivo exposures conducted. However, the results obtained from human primary 3D cell culture methods are applicable to human health outcomes. This model represents a physiologically relevant model of the human airway, comprising basal cells, mucus-secreting goblet cells, and ciliated cells, showing cilia beating and Mucociliary clearance [38, 39]. This model has been employed in studies to assess the respiratory toxicity of inhaled drugs targeting respiratory diseases [38] and has shown to express xenobiotic metabolism enzymes [40]. The single NP particle size selected does not represent the entire spectrum of nanoplastic particles that become aerosolized, but the 50 nm size is a common model particle size that may have higher uptake in cells [41-43].”

Comment 2: Line 42. We are already in 2025, so please amend the sentence.
Response 2: Thank you for the feedback! We have amended the line in 41-42 to now read “This figure is expected to triple by the year 2050 [2,3].”

Comment 3: Line 63. In vitro and in vivo do in italics.

Response 3: Thank you for this comment. We have updated the text on line 63 to have in vitro and in vivo in italics.

Comment 4: Line 212. Define SEM before use it.
Response 4: We thank the reviewer for catching this; we have updated the Statistics Section (2.8) to describe SEM before it is used in the Results section.

Comment 5: Please revise this article and include in your study: doi: 10.1016/j.intimp.2024.113921.
Response 5: Thank you for the feedback! We have included this as a reference for our paper.

Reviewer 3 Report

In this manuscript, author investigated the effects of acute NP exposure on a fully differentiated 3D human airway epithelial model derived from 14 healthy donors. Most of results were interesting to understand the e the potential for aerosolized NPs to induce oxidative and inflammatory stress, raising concerns about their long-term impact on respiratory health and redox regulation. The data are comprehensive and well presented in the figures, and the experimental approaches appear sound. However, the manuscript needs minor modifications to be considered by Antioxidants.

  • The impacts of aerosolized NPs reported in previous studies should be further described into Introduction. Ex) Characterization of changes in global genes expression in the lung of ICR mice in response to the inflammation and fibrosis induced by polystyrene nanoplastics inhalation. Jin YJ, Kim JE, Roh YJ, Song HJ, Seol A, Park J, Lim Y, Seo S, Hwang DY. Toxicol Res. 2023 May 22;39(4):1-25.
  • Why did you select doses of aerosolized NPs? You should describe a reason into Result or Discussion.
  • Author should add the limitation of this study and further study in Discussion or Conclusion section.
  • All number should be separated unit except % and o Also, unit should be described same pattern.
  • Also, all abbreviation should be fully described when it firstly appeared. Also, this description should be not repeated in text.
  • References should be corrected according to journal guideline.
  • In Figure 1, the statistical significance must be indicated
  • In Figure 2 legend, “**, ***, ****” should be deleted because they were not used in the figure.
  • In Figure 3 legend, “** and ****” should be described because they were used in the figure.

Author Response

Comment 1: The impacts of aerosolized NPs reported in previous studies should be further described into Introduction. Ex) Characterization of changes in global genes expression in the lung of ICR mice in response to the inflammation and fibrosis induced by polystyrene nanoplastics inhalation. Jin YJ, Kim JE, Roh YJ, Song HJ, Seol A, Park J, Lim Y, Seo S, Hwang DY. Toxicol Res. 2023 May 22;39(4):1-25.
Response 1: We thank the reviewers for this feedback. We have added this to our references and expanded on our Introduction section to now include “One study examined aerosolized PS particles at multiple concentrations for two weeks; the mice showed significant increases in inflammatory cytokines, histopathological changes, and the number of immune cells collected in the bronchoalveolar lavage fluid [16].”

Comment 2: Why did you select doses of aerosolized NPs? You should describe a reason into Result or Discussion.
Response 2: Thank you for the feedback! We have updated our Discussion to now include “The single NP particle size selected does not represent the entire spectrum of nanoplastic particles that become aerosolized, but the 50 nm size is a common model particle size that may have higher uptake in cells (https://doi.org/10.1016/j.chemosphere.2024.143702, doi: 10.3390/ijms25094724, and https://doi.org/10.1016/j.scitotenv.2022.155621).” as part of our justification for the PS used and the limitations of the study.

Comment 3: Author should add the limitation of this study and further study in Discussion or Conclusion section.
Response 3: We thank the reviewer for this feedback. We have updated our Discussion section to include the limitations of this study: “We utilized 3D model of cultured primary human airway epithelial cells to investigate the toxicity of ANPs by exposing the epithelial cells to PS-NPs by aerosolization. This study is limited in the variability of NP particle sizes tested and the lack of in vivo exposures conducted. However, the results obtained from human primary 3D cell culture methods are applicable to human health outcomes. This model represents a physiologically relevant model of the human airway, comprising basal cells, mucus-secreting goblet cells, and ciliated cells, showing cilia beating and Mucociliary clearance [38, 39]. This model has been employed in studies to assess the respiratory toxicity of inhaled drugs targeting respiratory diseases [38] and has shown to express xenobiotic metabolism enzymes [40]. The single NP particle size selected does not represent the entire spectrum of nanoplastic particles that become aerosolized, but the 50 nm size is a common model particle size that may have higher uptake in cells [41-43].”

Comment 4: All number should be separated unit except % and o Also, unit should be described same pattern.
Response 4: We thank the reviewer for this feedback. We have amended the text to have units separated from the values, except in the case of %s.Also, all abbreviation should be fully described when it firstly appeared.

Comment 5: Also, this description should be not repeated in text.
Response 5: Thank you for the feedback; we have amended the text so that all the abbreviations are described before their use in text, and the descriptions are not repeated.

Comment 6: References should be corrected according to journal guideline.
Response 6: Thank you for the feedback; we have corrected the references according to the journal guideline.

Comment 7: In Figure 1, the statistical significance must be indicated
Response 7: Thank you for the feedback! We have fixed Figure 1 to indicate statistical significance.

Comment 8: In Figure 2 legend, “**, ***, ****” should be deleted because they were not used in the figure.
Response 8: Thank you for this feedback; the Figure 2 Legend now only contains the statistical notation that is used within the figure.

Comment 9: In Figure 3 legend, “** and ****” should be described because they were used in the figure.
Response 9: Thank you for the comment! We have updated the Figure 3 legend to appropriately include all statistical notations.

Reviewer 4 Report

The paper “Acute Exposure to Aerosolized Nanoplastics Modulates Redox-2 Linked Immune Responses in Human Airway Epithelium” submitted to me for review is interesting given the ubiquity of micro- and nanoplastics (MPs and NPs) in the environment. Nanoparticles are a serious pollution problem that has been associated with adverse effects on human health, including inflammation and respiratory damage.

The work thoroughly aims to solve the imposed goals. The authors have clearly discussed the obtained results. However, I have some comments:

  1. The acquisition of primary human airway epithelial cells is not sufficiently clear to me. Whether they were purchased or obtained from the patient. However, there is no information about the consents of the Ethics Committees.

  1. Under Figure 1. There is information about statistical significance. However, there are no statistical significances in 4 graphs, is this a mistake or were there really no statistical significances anywhere? Even isoproterenol did not give statistical significances, so why was it used as a positive control?

  1. In Figure 2 there should be TGF-β, not b

Under Figure 1. There is information about statistical significance. However, there are no statistical significances in 4 graphs, is this a mistake or were there really no statistical significances anywhere? Even isoproterenol did not give statistical significances, so why was it used as a positive control?

In Figure 2 there should be TGF-β, not b

Author Response

Comment 1: Is the research design appropriate and are the methods adequately described? No. The acquisition of primary human airway epithelial cells is not sufficiently clear to me. Whether they were purchased or obtained from the patient. However, there is no information about the consents of the Ethics Committees.

Response 1: Thank you for the feedback! We did not collect the cells ourselves. The MucilAir system is constructed by Epithelix, where they handle the patient selection and sample collection. These are commercially available systems made with ethical approval and donor consent (https://www.epithelix.com/products/mucilair).
We have amended the text to now read: “A 3D model of cultured primary human airway epithelial cells was constructed from primary cells taken from 14 donors, acquired and provided by Epithelix with ethical approval and donor consent (MucilAir™; Epithelix; Geneva, Switzerland).”

Comment 2: Is it necessary to include study limitations in the discussion? Yes
Response 2: Thank you for the feedback! We have updated our Discussion to include the limitations of this study: “We utilized 3D model of cultured primary human airway epithelial cells to investigate the toxicity of ANPs by exposing the epithelial cells to PS-NPs by aerosolization. This study is limited in the variability of NP particle sizes tested and the lack of in vivo exposures conducted. However, the results obtained from human primary 3D cell culture methods are applicable to human health outcomes. This model represents a physiologically relevant model of the human airway, comprising basal cells, mucus-secreting goblet cells, and ciliated cells, showing cilia beating and Mucociliary clearance [38, 39]. This model has been employed in studies to assess the respiratory toxicity of inhaled drugs targeting respiratory diseases [38] and has shown to express xenobiotic metabolism enzymes [40]. The single NP particle size selected does not represent the entire spectrum of nanoplastic particles that become aerosolized, but the 50 nm size is a common model particle size that may have higher uptake in cells [41-43].”

Comment 3: Under Figure 1. There is information about statistical significance. However, there are no statistical significances in 4 graphs, is this a mistake or were there really no statistical significances anywhere? Even isoproterenol did not give statistical significances, so why was it used as a positive control?
Response 3: We thank the reviewer for their feedback! We have updated Figure 1 to indicate statistical significances.

Comment 4: In Figure 2 there should be TGF-β, not b

Response 4: Thank you for the feedback; we have corrected it as TGF-β.

Round 2

Reviewer 1 Report

The authors have addressed my comments with correction and additional description. I don`t have additional questions. 

No additional comments. 

Reviewer 4 Report

I have no comments anymore, and the changes made are satisfactory to me

I have no comments anymore, and the changes made are satisfactory to me